# *TUBB3* M323V Syndrome Presents with Infantile Nystagmus

**DOI:** 10.3390/genes12040575

**Published:** 2021-04-15

**Authors:** Soohwa Jin, Sung-Eun Park, Dongju Won, Seung-Tae Lee, Sueng-Han Han, Jinu Han

**Affiliations:** 1Department of Opthalmology, Yonsei University College of Medicine, Seoul 03722, Korea; 2015191084@yonsei.ac.kr; 2Department of Ophthalmology, Institute of Vision Research, Severance Hospital, Yonsei University College of Medicine, Seoul 03722, Korea; separk@yuhs.ac (S.-E.P.); shhan222@yuhs.ac (S.-H.H.); 3Department of Laboratory Medicine, Severance Hospital, Yonsei University College of Medicine, Seoul 03722, Korea; wdjbabo@yuhs.ac (D.W.); lee.st@yuhs.ac (S.-T.L.); 4Department of Ophthalmology, Institute of Vision Research, Gangnam Severance Hospital, Yonsei University College of Medicine, Seoul 06273, Korea

**Keywords:** infantile nystagmus, *TUBB3*, congenital fibrosis of the extraocular muscle, CFEOM3, tubulinopathy

## Abstract

Variants in the *TUBB3* gene, one of the tubulin-encoding genes, are known to cause congenital fibrosis of the extraocular muscles type 3 and/or malformations of cortical development. Herein, we report a case of a 6-month-old infant with c.967A>G:p.(M323V) variant in the *TUBB3* gene, who had only infantile nystagmus without other ophthalmological abnormalities. Subsequent brain magnetic resonance imaging (MRI) revealed cortical dysplasia. Neurological examinations did not reveal gross or fine motor delay, which are inconsistent with the clinical characteristics of patients with the M323V syndrome reported so far. A protein modeling showed that the M323V mutation in the *TUBB3* gene interferes with αβ heterodimer formation with the *TUBA1A* gene. This report emphasizes the importance of considering *TUBB3* and *TUBA1A* tubulinopathy in infantile nystagmus. A brain MRI should also be considered for these patients, although in the absence of other neurologic signs or symptoms.

## 1. Introduction

Infantile nystagmus syndrome is a genetically heterogeneous disorder in which an involuntary oscillation of the eyes begins within the first 6 months of life [1]. The oscillations usually start at 2 to 3 months of age when motor and visual functions develop and persist throughout life [2]. The prevalence of infantile nystagmus syndrome was estimated from 1 in 3000 to 1 in 1000 [3,4]. Infantile nystagmus can be idiopathic or associated with other ocular diseases, such as retinal disease, albinism, low vision, or loss of vision [5,6,7,8,9,10,11,12]. It can also occur as a common presenting sign of many neurologic and systemic diseases. It is noteworthy that nystagmus has psychological and social effects on children and their parents [13].

An ophthalmic examination involves careful observation of the nystagmus waveform, frequency, amplitude, direction, and the plane of oscillation, and the presence or absence of a null point [14]. Clinical workups, including optical coherence tomography, visual evoked potential, electroretinography (ERG), and genetic testing, are used to differentiate underlying causes of infantile nystagmus. As next-generation sequencing (NGS) technology has enabled us to examine multiple causative genes simultaneously, it is now used as a front-line diagnostic tool in infantile nystagmus patients [1,15].

Tubulin is a basic structural protein of microtubules, which play many instrumental roles, such as mitosis, axonal guidance, and neuronal migration during the development of the nervous system [16,17]. Therefore, mutations in tubulin genes can alter the normal function and structure of microtubules, leading to severe brain malformations, which are called tubulinopathies [18]. Among tubulin encoding genes, mutations in the *TUBB3* gene (OMIM #602661), encoding a class III β-tubulin, have been reported to cause two distinct congenital neuro-developmental pathologies: isolated or syndromic congenital fibrosis of the extraocular muscles type 3 (CFEOM3), or malformations of cortical development (MCD) [19]. CFEOM3 is a congenital, nonprogressive, oculomotor disorder that is characterized by variable deficits of vertical or horizontal eye movements and variable ptosis [20]. The limitation of eye movement in CFEOM3 patients is so various that they range from no or mild to severe ophthalmoplegia and may also show a unilateral or asymmetric presentation [21]. Interestingly, a recent study showed that the *TUBB3* gene variant could cause congenital monocular elevation deficiency [22]. MCD includes lissencephaly (agyria–pachygyria), polymicrogyria or polymicrogyria-like cortical dysplasia, and cortical gyral simplification. *TUBB3* mutations also affect the subcortical regions, generating dysplasia in the corpus callosum, cerebellar vermis, brainstem, basal ganglia, and cerebellum.

Herein, we report a case of c.967A>G:p.(M323V) variant in the *TUBB3* gene found in a male infant who had only infantile nystagmus without CFEOM.

## 2. Case Report

A 6-month-old male infant presented to our clinic with infantile nystagmus. The patient was born at full term (37 weeks), weighing 3.18 kg at the time of birth, after a normal pregnancy and delivery. He was the only child between non-consanguineous Korean parents, and neonatal and perinatal insults were not noted. His family history was also unremarkable.

On initial examination, he could not fix his eyes on an object and follow, and 1–2-Hz pendular nystagmus was noted. Cycloplegic refraction showed +sph1.50 in the right eye and +sph2.00 in the left eye. He had neither eye-poking signs nor photoaversion. Dilated fundus examination showed normal foveal reflex and normal optic disc at the posterior pole. An extraocular motility test showed a full range of motion. The neurological examination was also unremarkable. Targeted NGS revealed a heterozygous missense c.967A>G:p.(M323V) variant [Chr16(GRCh37):g.90001826A>G] in the *TUBB3* gene (NM_006086.4). This variant is absent in the population databases, such as the Genome Aggregation Database (gnomAD), the 1000 Genomes Project, and the Korean Reference Genome Database. This genomic position is highly conserved (phastCons: 1.00 and phyloP: 4.64). The M323 residue is located in exon 4 and conserved across β-tubulin isotypes from chicken to humans. Multiple lines of computational evidence support a deleterious effect of this variant (CADD: 25.0, FATHMM: 0.992). It was previously reported as pathogenic in ClinVar (RCV000023202.4). A de novo mutation was confirmed through segregation analysis (Figure 1A). This variant was classified as pathogenic (PS2, PM1, PM2, PP3, and PP5) according to the guideline of the American College of Medical Genetics [23].

Subsequently, brain magnetic resonance imaging (MRI) was performed, and it revealed an asymmetric configuration and size of caudate nuclei and asymmetric configurations of lateral ventricles, occipital lobes, and corpus callosum, which are consistent with cortical dysplasia (Figure 1B,C). Repeated examination of extraocular motility had shown full duction and version until the age of 23 months (Figure 1D), and 2-Hz left-beating jerk nystagmus and intermittent head nodding were observed. A non-sedated hand-held ERG test (RETeval, LKC Technologies, Gaithersburg, MD, USA) using skin electrodes was performed for the diagnosis of retinal dysfunctions associated with nystagmus. The scotopic response was normal, but the result was inconclusive due to poor patient cooperation (Figure 2). Neurological examinations showed no gross or fine motor delays. He could sit, walk, and even run without support, but mild intellectual disability and mild language delay were noted.

## 3. Discussion

Our case demonstrates that the *TUBB3* M323V syndrome causes infantile nystagmus without CFEOM. In a previous study, the heterozygous c.967A>G:p.(M323V) *TUBB3* variant caused nystagmus phenotypes without CFEOM in two patients in the same family (father and son) [24]. Table 1 summarizes the clinical findings in previous cases of infantile nystagmus with the *TUBB3* variants, as well as the case described in this report. Previously reported patients with the M323V syndrome had no CFEOM phenotype, but patients with G71R and G98S syndrome showed both CFEOM and infantile nystagmus [19,24]. Among seven patients, including our case, five patients had horizontal nystagmus, and the other two patients showed multidirectional and rotary nystagmus, respectively.

Our report demonstrates that the *TUBB3* gene should be considered as a causative gene for infantile nystagmus. The heterozygous missense mutation c.967A>G:p.(M323V) is located at the intermediate domain (residues 230–371) in a class III β-tubulin (Figure 3B), which engages in heterodimer stability and longitudinal and lateral interactions [25]. In a protein model using UCSF ChimeraX [26], a p.M323V is predicted to cause a clash between TUBB3:p.V323 and TUBA1A:p.Y210, which may affect the stability of the heterodimer (Figure 3A). The *TUBB3*:M323 residue is the interaction site with *TUBA1A* when forming heterodimers. The proposed mechanism of nystagmus phenotype in M323V syndrome is an impaired capacity to form αβ tubulin heterodimers, not through the independent mechanism of GTP-binding. The present case resembles *TUBA1A*-associated tubulinopathy, rather than classic *TUBB3* CFEOM3, where nystagmus was present in 3/29 (10.3%) cases, and no CFEOM phenotypes were observed [27].

Optic nerve hypoplasia has been reported with *TUBA1A*, *TUBB2B*, and *TUBA8* mutations, suggesting that tubulin gene mutations, in general, can cause optic nerve hypoplasia [28,29,30]. The possibility that optic nerve hypoplasia is the cause of nystagmus cannot be excluded. Although optic nerve hypoplasia has been reported in tubulinopathies, there were no reports of infantile nystagmus in patients with optic nerve hypoplasia who had *TUBB3* mutations. Moreover, a dilated fundus examination and brain MRI did not reveal optic nerve hypoplasia in our case. *TUBB3* has widespread expression in the retinal ganglion cells, amacrine cells, horizontal process, and cone photoreceptors [31]. The neuronal circuit of direction-selective retinal cells may be disrupted due to *TUBB3* mutation [32]. Because we could not obtain optical coherence tomography, it is possible that a mild degree of foveal hypoplasia or retinal dystrophy co-exists. However, our targeted panel included 429 genes associated with inherited retinal diseases and infantile nystagmus syndrome, so we can exclude those possibilities. Further research is needed to determine whether the cause of nystagmus is due to cortical, cerebellar, or retinal origin [33].

In conclusion, our case report shows that infantile nystagmus can arise without CFEOM owing to the *TUBB3* variant. Therefore, pediatric ophthalmologists should keep in mind that the clinical features of the *TUBB3* syndrome are so diverse that only nystagmus could appear as the main presenting sign. We also thought that *TUBB3* and *TUBA1A* genes should be included in the targeted panel of infantile nystagmus. In general, a brain MRI has a low diagnostic yield for patients with infantile nystagmus in the absence of other neurologic signs or symptoms [34]. However, as in this case, an accurate molecular diagnosis will enable clinicians to determine whether a brain MRI is necessary or not. Additionally, collaborations with multiple specialties may facilitate the appropriate management in such cases.

## Figures and Tables

**Figure 1 genes-12-00575-f001:**
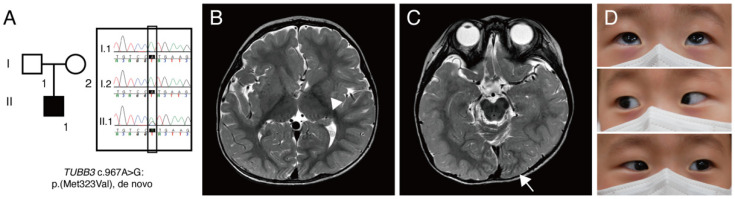
(**A**) A pedigree of patient reported in this study. Square, male; round, female; black coloring, affected individual. Targeted next-generation sequencing showed *TUBB3* c.967A>G:p.(M323V) variant. Sanger sequencing confirmed that this variant is a de novo mutation. (**B**) Brain magnetic resonance imaging showing cortical dysplasia. T2-weighted images without contrast revealed an asymmetric caudate nucleus (arrowhead) and globular shape of both basal ganglia and thalamus. (**C**) Axial T2-weighted image showing an asymmetric configuration of an occipital lobe (arrow) and abnormal cerebellar vermian foldings. (**D**) Pictures of an extraocular motility examination showing a full range of motion.

**Figure 2 genes-12-00575-f002:**
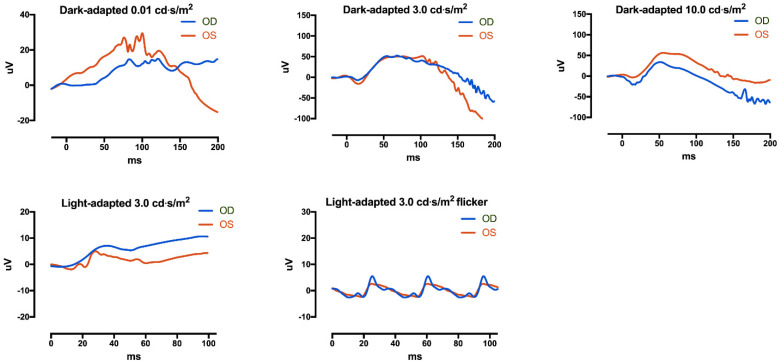
Electroretinography (ERG) was performed with skin electrodes. (**Top**) Dark-adapted 0.01 ERG, dark-adapted 3.0 ERG, and dark-adapted 10.0 ERG showed relatively normal waveforms, but the patient was very uncooperative during examinations. (**Bottom**) Light-adapted 3.0 ERG 3.0 flicker was obtained. Photopic ERG responses seemed to be reduced, but the result was inconclusive due to poor cooperation. The flash strength unit is cd·s/m^2^.

**Figure 3 genes-12-00575-f003:**
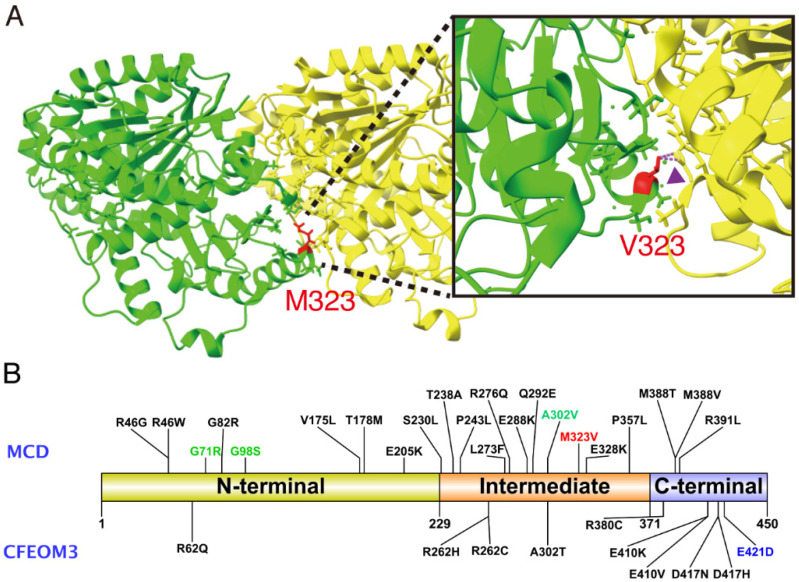
(**A**) A protein model with UCSF ChimeraX, showing the unstable formation of αβ tubulin heterodimer in M323V syndrome. Green, class III β-tubulin encoded by *TUBB3*; Yellow, α tubulin encoded by *TUB1A1*. A clash occurred between *TUBB3*:p.V323 and TUBA1A:p.Y210 (arrowhead). (**B**) Schematic diagram of deleterious variants in *TUBB3* functional domains. A total of thirty-two missense variants, including p.M323V, have been reported until recently. Variants associated with malformations of cortical development (MCD) with or without congenital fibrosis of the extraocular muscles type 3 (CFEOM3) are marked above, and variants only representing CFEOM3 are marked at the bottom. Most variants in the N-terminal and the intermediate domain cause MCD, and missense variants in the C-terminal cause either MCD or CFEOM3 phenotypes. A red word indicates the variant in this study. Green words denote previously reported variants associated with infantile nystagmus, and a blue word indicates a variant associated with monocular elevation deficiency.

**Table 1 genes-12-00575-t001:** Literature review of clinical characteristics in infantile nystagmus patients with the *TUBB3* variants.

Clinical Characteristics		Patient 1	Patient 2	Patient 3	Patient 4	Patient 5	Patient 6	Patient 7
*TUBB3* variant		p.M323V	p.M323V	p.A302V (homo)	p.G71R	p.G71R	p.G98S	p.M323V
Inheritance pattern		AD	AD	Isolated (homo)	Isolated	Isolated	Isolated	Isolated
Age		36 years	2 years	1 year	5 years	9 years	2 years	6 months
Gender		Male	Male	Female	Female	Male	Female	Male
Ethnicity		NA	NA	NA	European	European	European	Korean
OFC		3rd p	25th p	3rd p	NA	NA	NA	3-50th p
Motor delay		Hypotonia	Hypotonia	Hypotonia	Hypotonia	Hypotonia	Hypotonia	Absent
Cognitive function		Severe ID	LD	NA	ID	ID	ID	ID, LD
Epilepsy		Absent	Absent	Absent	NA	NA	NA	Absent
CFEOM		No	No	No	Yes	Yes	Yes	No
Nystagmus		Horizontal nystagmus	Horizontal nystagmus	Multidirectional nystagmus	Rotary nystagmus	Horizontal nystagmus	Horizontal nystagmus	Horizontal nystagmus
Cortical dysgenesis		Gyral disorganization	Gyral disorganization	Gyral disorganization	Gyral disorganization	Gyral disorganization	Gyral disorganization	Gyral disorganization
Cerebellum	vermis	Dysplastic	Dysplastic	Dysplastic	Dysplastic	Dysplastic	Dysplastic	Dysplastic
	Hemisphere	Dysplastic	Normal	Normal	Normal	Normal	Normal	Normal
Brainstem		Hypoplastic	Hypoplastic	Hypoplastic	Hypoplastic	Hypoplastic	Hypoplastic	Normal
Corpus callosum		Thin	Thin	Thin	Thin	Thin	Thin	Asymmetric
Basal ganglia		Hypertrophic/mild fusion	Hypertrophic/mild fusion	Fusion caudate/ putamen	Hypertrophic/ fusion	Hypertrophic/ fusion	Hypertrophic/ fusion	Asymmetric
Literatures		Poirier et al. Hum Mol Genet (2010)	Poirier et al. Hum Mol Genet (2010)	Poirier et al. Hum Mol Genet (2010)	Whitman et al. Am J Med Genet A. (2016)	Whitman et al. Am J Med Genet A. (2016)	Whitman et al. Am J Med Genet A. (2016)	This study

Abbreviations: AD, autosomal dominant; CFEOM, congenital fibrosis of the extraocular muscle; homo, homozygous; ID, intellectual disability; LD, language delay; NA, not available; OFC, occipitofrontal circumference; p, percentile.

## Data Availability

The data presented in this study is contained within the article.

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
