# Peer review of "TUBB3 M323V Syndrome Presents with Infantile Nystagmus"

_genes, 2021, doi:10.3390/genes12040575_

Round 1

Reviewer 1 Report

This is an interesting study presented by Jin et al. highlighting that the TUBB3 M323V variant can present with infantile nystagmus. The study by Poirier et al. (2010) also shows that the patients with this variant can have nystagmus. Taken together, there is now good evidence of the association of nystagmus due to this variant. Surprisingly patients with this variant do not have the CFEOM phenotype as described in this study and the previous one by Poirier et al. 2010

There are some minor suggestions:

1. The authors show that there is abnormal vermian folds in figure 1C. However in table 1 state that the vermis is normal. Unlike other forms of INS could the aetiology of nystagmus be related to the cortical or vermian changes? 

2. INS requires the presence of an accelerating slow phase - was this seen in the patient described in the case report?

3. Is there any evidence of an abnormal optic nerve (such as optic nerve hypoplasia) which has been described in TUBA8 (Abdollahi, M.R et al. Mutation of the variant alpha-tubulin TUBA8 results in polymicrogyria with optic nerve hypoplasia AJHG 2009), TUBA1A (Aiken J et al. The alpha-Tubulin gene TUBA1A in Brain Development: A Key Ingredient in the Neuronal Isotype Blend. J Dev Biol 2017)  and TUBB3 (Thomas MG et al. Optic Nerve Head and Retinal Ab-normalities Associated with Congen-ital Fibrosis of Extraocular Muscles. Int. J. Mol. Sci. 2021). It is recommended that the authors discuss these previously reported findings in relation to their case since it could be that tublin gene mutations in general can cause optic nerve hypoplasia. If there is evidence of optic nerve hypoplasia it could also explain the aetiology of the nystagmus. 

4. In terms of terminology, it might be better to call this just infantile nystagmus rather than infantile nystagmus syndrome (unless evidenced by eye movements and an accelerating slow phase).

5. Overall, the authors provide a convincing story about nystagmus association with this variant this is further backed by previous literature. However there should be more of a discussion about the nystagmus including mechanisms (optic nerve hypoplasia vs cerebellar mechanism vs cortical dysplasia).

Reviewer 2 Report

The authors have examined an isolated case of infantile nystagmus syndrome without CFEOM and identified a heterozygous mutation in TUBB3 gene. The ophthalmological examination and gene analysis technique which authors applied in this study were regular methods, and adequate to clarifies the information we need. This is an interesting case study, however there’s some unclear information which I have listed below too. I recommend to accept this case study with subject to major revisions. Following are my specific comments;

  1. In line 29 “INS can be idiopathic or associated to other ocular diseases, such as retinal disease, albinism, low vision or loss of vision”. Please mention the reference for retinal diseases and albinism.
  2. In line 29 “INS can be idiopathic or associated ‘to’ other ocular diseases….” ‘with’ instead of ‘to’.
  3. Mutation examination: I would advise authors to add electropherogram or sanger sequencing results for both parents and affected child which would indicate the segregation of mutation.
  4. Author should also mention the exon number for this reported mutation.
  5. Fundus examination: Sometimes, infantile nystagmus itself associated with other retinal dystrophies. I recommend authors to add fundus pictures and /or OCT pictures of patient or patient in comparison with an unaffected parent(s).
  6. In section 3 of Discussion; as this mutation has already been reported earlier by Poirier et al (2010). Please highlight how this study is significantly different from the already reported study for the same mutation. I invite authors to discuss more on clinical phenotypes of patient in comparison with the clinical phenotypes reported by Poirier et al. 2010.
  7. Comparison of phenotypes in tabular form is informative. I recommend authors to make the table more informative; add study done by DOI: 1016/j.cell.2009.12.011; add Inheritance pattern (AD, AR, Isolated) column.

Round 2

Reviewer 2 Report

The authors have addressed all my comments effectively. The manuscript now looks good for publication.